# Healthcare professional and commissioners' perspectives on the factors facilitating and hindering the implementation of digital tools for self-management of long-term conditions within UK healthcare pathways

**James P. Gavin**[1]*, **Paul Clarkson**[1,2], **Paul E. Muckelt**[1], **Rachael Eckford**[1], **Euan Sadler**[1], **Suzanne McDonough**[1,3], **Mary Barker**[1,4]

1 School of Health Sciences, University of Southampton, Southampton, Hampshire, United Kingdom, 2 NIHR Applied Research Collaboration Wessex, Southampton, Hampshire, United Kingdom, 3 School of Physiotherapy, RCSI University of Medicine and Health Sciences, Dublin, Ireland, 4 Faculty of Medicine, University of Southampton, Southampton, Hampshire, United Kingdom

* J.P.Gavin@soton.ac.uk

**Editor:** Sebastian Suarez Fuller, University of Oxford Nuffield Department of Clinical Medicine: University of Oxford Nuffield Department of Medicine, UNITED KINGDOM OF GREAT BRITAIN AND NORTHERN IRELAND

## Abstract

Physical activity is important in the self-management of long-term conditions (LTCs). However, implementing physical activity into clinical practice is challenging, due to complex barriers including access to programmes, time pressures, and transport costs, for people with comorbidities, managing multiple responsibilities. Various digital tools exist to overcome these barriers and support wide-scale implementation to help people stay physically active. We explored the experiences, needs and preferences of healthcare professionals and commissioners, regarding the use of digital tools to support people with LTCs to self-manage using physical activity. This included barriers and facilitators to implementing digital tools to support people with LTCs in NHS settings. Semi-structured interviews were conducted (April 2021 to January 2022) in Wessex, southern England, UK. Purposive sampling was used to recruit general practitioners and healthcare professionals, and convenience sampling to recruit commissioners (n = 15). Transcripts were coded to develop conceptual themes allowing comparisons between and among perspectives, with the Normalisation Process Theory (NPT)'s four constructs used to aid interpretation. Results showed that most digital tools supporting physical activity for LTCs, are not well implemented clinically. Current digital tools were seen to lack condition-specificity, usability/acceptability evidence-base, and voluntary sector involvement (i.e., NPT: coherence or 'making sense'). Healthcare professionals and commissioners were unlikely to engage with use of digital tools unless they were integrated into health service IT systems and professional networks (i.e., NPT: cognitive participation), or adaptable to the digital literacy levels of service users and staff (i.e., NPT: collective action–needs for implementation). In practice, this meant being technically, easy to use and culturally accessible (i.e., NPT: collective action–promoting healthcare work). COVID-19 changed professional attitudes towards digital tools, in that they saw them being viable, feasible and critical options in a way they had not done before

**Data Availability Statement:** All relevant data are within the manuscript and its Supporting Information files. Interview transcripts and coding/thematic analysis document (including fieldnotes) can be accessed upon request from the research team.

**Funding:** This study was funded by the National Institute for Health Research (NIHR) Applied Research Collaboration (ARC) Wessex. The views expressed in this publication are those of the author(s) and not necessarily those of the National Institute for Health and Care Research or the Department of Health and Social Care (https://www.arc-wx.nihr.ac.uk/). The funders had no role in study design, data collection and analysis, decision to publish, or preparation of the manuscript.

**Competing interests:** The authors have declared that no competing interests exist.

the pandemic. Implementation was also influenced by endorsement and trustworthiness enhancing the perception of them as secure and evidence-based (i.e., NPT: reflective monitoring). Our findings highlight that consideration must be given to ensuring that digital tools are accessible to both healthcare professionals and patients, have usability/acceptability, and are adaptable to specific LTCs. To promote clinical engagement, digital tools must be evidence-based, endorsed by professional networks, and integrated into existing health systems. Digital literacy of patients and professionals is also crucial for cross-service implementation.

## Introduction

Physical activity is important in the management and prevention of long-term conditions (LTCs), specifically to improve symptoms, support individuals in remaining active, and mitigating future health problems [1]. Existing systematic reviews and guidelines make clear the benefits of physical activity in managing symptoms across LTCs, in addition to preventing complications and preserving function [2–5]. An estimated 15 million people in the UK live with one or more LTCs, which includes both mental and physical health conditions that cannot be cured, but can be managed with therapy and/or medication [6]. Examples of LTCs include cardiovascular disease, stroke, chronic obstructive pulmonary disease (COPD), and depression. Self-management is fundamental in the day-to-day care of LTCs; it can support the individual to apply strategies to manage their symptoms and undertake daily activities [7, 8], whilst experiencing the lifestyle changes associated with their LTC. Healthcare professionals are the pivotal, first contacts, to support patients in managing their own conditions, being active in their self-management, self-care, and decision-making. However, healthcare professionals have little direct involvement in the self-management of LTCs on a regular basis, when people with LTCs return to their communities. Individuals instead manage their own health, but with limited advice and education from healthcare professionals [9]. Regular physical activity is one such general health behaviour advocated by health professionals for the management of LTCs. However, in practice, the adoption of physical activity for the self-management of LTCs is often overlooked [2], with many healthcare professionals lacking knowledge of national physical activity guidelines [3]. Clinical endorsement is crucial in promoting physical activity [10], and in particular, the growing use of digital tools in self-managing LTCs [11–13].

Approximately 27.5% (1.4 billion) of the adult population worldwide do not meet the World Health Organisation's (WHO) recommended levels for physical activity [1, 14], yet physical activity is commonly adopted in interventions to support LTCs [2, 3]. Translating physical activity into clinical practice is an ongoing challenge due to multifactorial and complex barriers to implementation. Significant barriers include: healthcare professionals not always having the skills and knowledge to support people with LTCs (particularly those who are fearful that physical activity may exacerbate their conditions [15–17]); transport costs; comorbidities, and; competing family obligations (e.g. caring for others). These are in addition to other barriers, such as lack of awareness and low social support, preventing regular engagement in face-to-face self-management programmes to promote physical activity behaviour change [12, 18, 19]. Identifying safe and adaptable healthcare strategies to facilitate the uptake and maintenance of regular daily activity in people with LTCs will be important in engaging them to maintain self-management. Arguably more crucial is the long-term maintenance of

changed behaviour, which ultimately leads to the majority of positive health outcomes including weight management, reduced cardiovascular risk and functional strength associated with physical activity [20].

Various digital and non-digital programmes and tools exist, including mobile applications, websites, and exercise referral schemes, to help support individuals to self-manage their LTCs and to become physically active. Little evidence currently exists to support the effectiveness of these programmes in helping individuals to remain physically active in the longer term [21]. This may partly be because, traditional self-management programmes are not always tailored to individual's needs, LTC(s) or home and community environments [22, 23]. This could explain why so few physical activity programmes have been implemented large-scale in people's communities over the last 20 years [24]. Lower socio-economic groups are particularly disadvantaged by standard self-management interventions, due to factors such as lack of transportation (including finances to support travel), caring need for dependent family members and, particularly for community-based interventions, a lack of health literacy [25, 26]. Digital tools have the potential to overcome barriers presented by travel, cost, and public accessibility, and support wide-scale implementation of programmes for physical activity maintenance [13, 27]. The uptake and effective integration within the UK National Health Service (NHS) has been historically poor.

The Normalisation Process Theory (NPT) is a framework that can be applied to understand and explain the processes preventing uptake, and effective integration, of interventions or innovations [28]. In this case, digital tools into national health services. It comprises four constructs: i) coherence (making sense of the intervention), ii) cognitive participation (enrolling individuals to engage with the practice), iii) collective action (enacting the practice) and iv) reflective monitoring (informal and formal appraisal of the practice). We adopted the NPT to help us understand facilitators supporting, or barriers inhibiting the implementation of digital tools, from the perspectives of healthcare professionals and commissioners using them. Our earlier scoping review [21] identified a range of digital tools to support the maintenance of physical activity for people with LTCs, their characteristics and their theoretical underpinnings.

The aims of the study reported in this paper were twofold:

- To explore the needs and preferences of healthcare professionals and service commissioners, with regard to using and recommending digital tools to support self-management for maintaining physical activity for people with LTCs;

- To explore barriers to and facilitators of implementation of digital tools by healthcare professionals and commissioners of services to support people with LTCs in NHS settings.

This paper reports on a qualitative exploration of barriers and enablers to implementation that addresses the aims above. Increased understanding of these will facilitate effective implementation of digital interventions into health and social care [29, 30].

## Materials and methods

### Study design

Maintenance Of physical acTivity beHaviour (MOTH) was a mixed methods research programme exploring digital and non-digital behaviour change interventions to support the maintenance of physical activity in adults with LTCs (ISRCTN: 16805986I). The programme contained three components exploring the role of digital interventions in supporting physical activity maintenance: i) a scoping review [21], ii) systematic review (PROSPERO

CRD42022299967), and ii) interviews with healthcare professionals and commissioners of services supporting people with LTCs in the NHS. The scoping review has been published [21]. This paper reports the findings from the interview component of this research and progresses a previous NIHR project to develop and assess the feasibility of digital tools for the self-management of joint pain. It is also part of a wider project to develop a digital health intervention to support the maintenance of physical activity in people with LTCs. The study reported in this paper was approved by the University of Southampton (ERGO ref: 60495.A2) and NHS HRA Research Ethics committees (IRAS ref: 288651).

## Sampling and recruitment

This qualitative, semi-structured interview study involved recruitment of general practitioners (GPs), other healthcare professionals specialising in LTCs (e.g., nurses, physiotherapists and occupational therapists) and service commissioners from what were previously known as Clinical Commissioning Groups (CCG) (as of 1$^{st}$ July 2022 Integrated Care Boards) in the Wessex region, southern England. Participants were recruited via Wessex CCGs, Solent NHS Trust, local Clinical Research Network (CRN) groups, and existing clinical academic networks (between 6 January and 11 December 2021).

GPs and LTC healthcare specialists were sampled purposively from amongst Wessex CRN and Solent NHS Trust staff purposively to reflect diversity in geographical location, gender, age and LTC specialism/role. Wessex CRN and NHS Trust administrators emailed their clinical and primary care networks with information about the study. Potential participants then responded directly to staff in the CRN, NHS Trust or research teams to arrange an interview. This has previously been effective in achieving maximum variation in demographic characteristics (gender, age, location, and LTC management experience) for GP interviews [31]. Convenience sampling was used to recruit service commissioners. They were emailed directly and all those who agreed to take part were interviewed. This approach was taken following previously reported difficulties in recruiting service commissioners to take part in research [32].

Participants had to be NHS commissioners of services, GPs or healthcare professionals, have experience in providing care for people with a LTCs, have recommended or used digital tools to support their LTC patients, be an English speaker, and aged ≥18 years. Written and verbal informed consent was obtained from all participants prior to interview, by either co-author PC or PM. Initially the researchers planned to interview a range of stakeholders; they were aiming for ten each of GPs, LTC specialists and commissioners. However, a sample size of nine has previously been used to achieve data saturation, which was defined as the point at which there no new themes are identified during data collection and analysis [33].

## Data collection

Interviews were conducted between April 2021 and January 2022, either in-person or online (via Microsoft Teams) by two researchers (PC and PM), with each interview lasting 45–65 minutes. Each audio-recorded interview focused on four topics: i) the participant's experiences of recommending or supporting a digital health intervention in practice; ii) their perceptions of what constituted a successful digital health intervention; iii) perceived barriers and enablers to digital health tools, which support physical activity in their setting; and iv) the influence of policy/guidelines on the implementation or recommendation of digital health interventions in their service(s) (S1 Fig). Basic socio-demographic information was also collected on gender, age and location of practice/commissioning area of each participant.

Three authors (PC, ES and SMcD) developed the semi-structured interview topic guide (S1 Fig), which was informed by the Consolidated Framework for Implementation Research

(CFIR) [34]. The CFIR helped ensure that the interviews produced information to help the authors understand the contextual factors for the implementation of digital tools in the NHS, specifically, in relation to the following: intervention characteristics including the evidence-base and cost; outer settings (e.g., patient needs and resources); inner settings (e.g., infrastructure and culture of NHS services); individuals including knowledge and beliefs; and the intervention process including planning and engaging. The interview guide was further refined through discussion and feedback within the research team (PC, SMcD and PM). Finally, content and face validity were checked by having the interview guide peer-reviewed by GP and LTC healthcare professional colleagues at the University of Southampton.

## Data analysis

Audio recordings of the interviews were transcribed verbatim using Microsoft Word, with participant's names and locations anonymised, and then imported to NVivo (release 1.6.1 [1137]) for thematic analysis. Firstly, analysis involved one member of the team (RE), reading the transcripts several times for familiarisation, before developing initial codes (themes). Secondly, the code clusters from NVivo, were used to construct a coding map. This helped organise the initial themes into higher order themes, and related subthemes developed from the interview data. Then the 'one sheet of paper method' was used outside NVivo. This involved one author (RE) i) visually mapping the themes and related subthemes, and then ii) relationships between themes and subthemes on one side of A4 paper, to iii) further develop the higher order themes [35]. This also offered insights into similarities and differences across healthcare professional and commissioner perspectives. This process was undertaken several times, to develop a number of iterations of the 'one sheet of paper' visually mapped, themes, which were developed in discussion with other co-authors (ES and JG). Finally, the findings from each theme were mapped to the four constructs of NPT [28] (Fig 1). A consensus on the final themes and related subthemes was produced from discussion with all authors.

## Results

Overall, 15 professionals were interviewed, comprising two commissioners (one in digital health), ten GPs, two nurses, and one physiotherapist. They were aged between 39 and 63 years and eight were women (Table 1). All participants had previously recommended a digital tool to support patients with LTCs to maintain physical activity, these included the following: mobile applications (or 'apps'), telehealth (e.g., blood pressure monitoring and online exercise classes), websites, artificial intelligence, online consultations, text and email communications, and algorithms (i.e., for prognostic monitoring). Digital tools were perceived by our healthcare professionals to range from apps to websites to Excel spreadsheets. Nine participants had used a digital tool as part of research, to deliver and evaluate the impact of the tool on LTC patients.

Our findings are presented and interpreted in relation to the NPT [28], and focus on factors that healthcare professionals perceived to support successful implementation of digital interventions in healthcare (Table 2).

## Coherence: Making sense of the digital tool

**Barrier 1: Condition-specific and accessible tools.**  Thirteen of the participants (eight GPs, both commissioners, two nurses and one physiotherapist) felt that one digital tool, such as a smartphone app for self-management of physical activity, could not possibly be designed to suit everyone. Participants explained that some basic advice would be the same, but for conditions such as COPD and diabetes, suitable modes and intensities of physical activity/exercise would be different. For example, people with COPD experience 'exercise' dyspnea (i.e.,

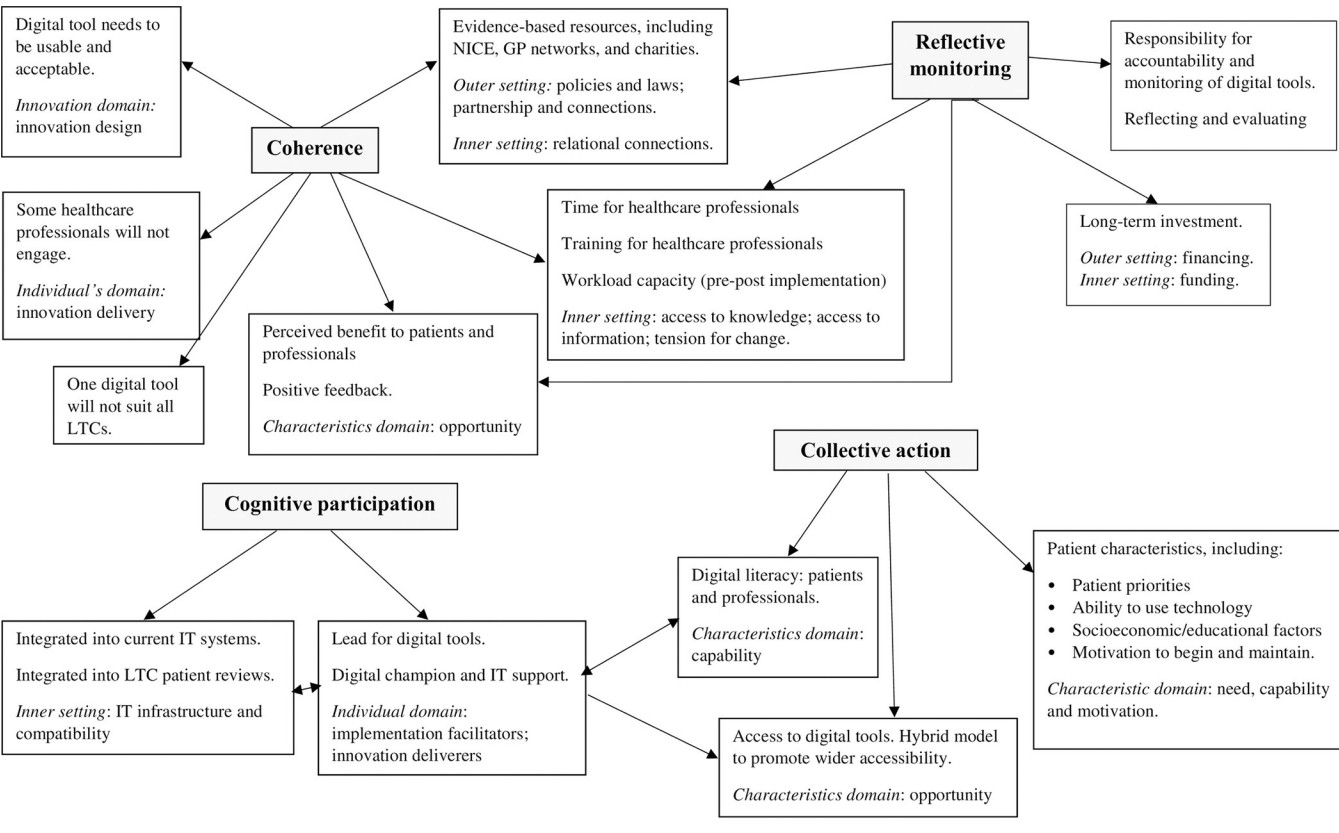

**Fig 1. Schematic map of conceptual themes.**

breathlessness) due to neuromechanical dissociation, which is a mismatch between increased respiratory workload and altered sensory feedback, leading to breathlessness on exertion, even during low-intensity physical activity.

*"We group some of them* [health conditions] *together because they are quite similar. In the practice, for example when we do cardiovascular review, those are all patients who have diabetes and/or hypertension and/or heart disease because generally the data for that is all the same. So, we group them together but if they have asthma as well that's not really that relevant and none of the things that you do for asthma maybe relevant."* GP, participant 13.

"*There will be common threads that you will probably find most health lifestyle Apps will recommend people altering their fat and sugar intake etc. but yes there will be big differences*" GP, participant 11.

A potential solution raised by some participants, was to have a 'generic' digital tool to promote physical activity maintenance, that could address the support needs of those with all major LTCs, but also be adaptable to the individual's specific condition(s) (e.g., cancer and chronic fatigue). For example, another GP commented:

"...*there are the more general ones that you have a good knowledge of and how they work locally. Then there are some condition-specific ones ... it's important to get the right patients through to those ones, and I think an online long term condition tool could hopefully help you signpost to the appropriate ones* [LTCs]." GP, participant 9.

Table 1. Participant characteristics.

| Participant number | Profession | Age (yrs) | Gender | Region | Types of LTCs supported | Digital tool(s) adopted |
|---|---|---|---|---|---|---|
| 1 | GP | 59 | Man | Southeast Hampshire, UK | Diabetes | App integrated into GP IT systems. |
| 2 | Commissioner | 39 | Woman | Dorset, UK | Hypertension COPD | Remote monitoring tool. Digital tool initially used but withdrawn due to a lack of continued patient and professional engagement. |
| 3 | GP | 41 | Man | South Hampshire, UK | Various | Home blood pressure monitoring tool; Excel spreadsheets for sorting and storing data; e-consult. |
| 4 | GP | 42 | Man | Southeast Dorset, UK | COPD and heart failure Eczema | Experience of using various digital tools in the past 5-years (across two hospital trusts). Including a digital tool measuring oxygen saturation, which informs GP intervention. Also, piloted an ECG device, which did not work in practice. Web-based package for patient education and self-management. |
| 5 | Practice Nurse | 59 | Women | Wiltshire | Various | Signposting to asthma UK and British Lung Foundation websites, and generic weight reduction and exercise programmes. |
| 6 | Consultant physiotherapist | 51 | Man | Hampshire | Various | Apps specific to long-COVID*, diabetes and COPD. * Useful aid, but not fully supported for self-management |
| 7 | Nurse | 63 | Woman | Hampshire | Dementia | Telecare, which included memory clocks; GPS trackers; automated medication reminders. |
| 8 | GP | n/d | Woman | Surrey | Various | Diabetes app; e-consult; NHS app. |
| 9 | GP | 36 | Man | Dorset | Various | Online blood pressure monitoring. |
| 10 | Commissioner | 42 | Woman | Hampshire & Isle of Wight | Cardiovascular disease | 'myHeart' and 'myHealth' apps for cardiac rehabilitation; long-COVID app. |
| 11 | GP | 33 | Man | Hampshire | Type II diabetes Chronic kidney disease Various | 'My Desmond' app. 'My Kidneys' app; online resources. NHS app. |
| 12 | GP | 43 | Man | Dorset | Various | Recommends mental health app; online blood pressure monitoring. |
| 13 | GP | 46 | Man | Buckinghamshire | Various | NHS app. |
| 14 | GP | 54 | Woman | Berkshire | Various | Online blood pressure monitoring (via text message); e-consult. |
| 15 | GP | 33 | Woman | Dorset | Various | 'Hypertension Plus' app; type II diabetes app. |

CCG, Clinical commissioning group; ECG, Electrocardiogram; GP, General practitioner; GPS, global positioning system; LTC, Long-term condition; NHS, National Health Service (UK). n/a, not disclosed.

This would allow the patient to add or change the LTCs, with which they were offered support, with guidance from their healthcare teams as appropriate, and to account for the processes of ageing and/or frailty. This would ultimately promote coherence for both patients and professionals.

**Facilitator 1: Usability and acceptability.** Participants commonly reported that for them to promote a digital tool with their patients, it would need to be usable and acceptable. In particular, whether a digital tool was easy to use and was accompanied by customer support (i.e., usability), and did what it claimed to do, whilst providing accurate and reliable data (i.e., acceptability). They often discussed apps and digital devices they had previously trialled, which had not worked, because they had either produced inaccurate data and/or were difficult to set-up. All GPs, one nurse and the physiotherapist, believed that a digital tool would also need to enable a patient to increase, or maintain their physical activity as part of their daily self-management, and accurate feedback to healthcare professionals. For example, one GP said:

**Table 2. Overview of themes to support the implementation of digital tools in healthcare for people living with long-term conditions.**

| Domain | Theme |
|---|---|
| **Coherence**: making sense of digital tools | **Facilitator** |
| | • Usability and acceptability |
| | • Evidence-based digital tools |
| | • Voluntary sector involvement |
| | **Barrier** |
| | • Engagement of healthcare professionals with tools |
| | • Professionals investing time in digital tools |
| | Condition-specific and accessible tools |
| **Cognitive participation**: enrolling and engaging individuals with new digital tools | **Facilitator** |
| | • GP network involvement |
| | • Digital leads or champions |
| | • Changing attitudes post-COVID-19 |
| | **Barrier** |
| | • Integration into current IT systems |
| **Collective action**: what needs to be done to enact new digital tools | **Barrier** |
| | • Digital literacy |
| | • Patient characteristics |
| **Reflective monitoring**: informal and formal appraisal of digital tools over time | **Facilitator** |
| | • Positive feedback from professionals and patients |
| | **Barrier** |
| | • Long-term investment |
| | • Accountability and monitoring |

Themes are interpreted in relation to the NPT domains [28] and from the perspectives of commissioners and healthcare professionals. GP, general practitioner.

> *"Just functionality. I think that applies globally, that applies to any apps. That's not unique to this. But any app that doesn't work, everyone gives up on very quickly."* GP, participant 8.

> "*So, something that's really intuitively easy to use like when you get an iPhone now and there's no instructions that come with it at all apart from one tiny bit of cardboard that says to turn it on or something. Then it's so well designed that you just know how to use it. It needs to be really well thought out and tested.*" GP, participant 3.

> "*. . .number two [priority] is that it's acceptable and welcomed by staff as well. . . usability has got to be key to making it, to get patients to adopt it and stick with it.*" GP, participant 12.

**Facilitator 2: Evidence-based digital tools.** All participants believed that to be adopted into practice and recommended to patients, any digital tool would need to have a clear evidence-base that they improve health outcomes. Healthcare professionals explained their commitment to patient care meant that they would not be able to recommend something that had not demonstrated effectiveness in supporting patients with LTCs, particularly in staying physically active. Most of the participants, referred to the National Institute for Health and Care Excellence (NICE) guidelines, as a reliable and trusted evidence-base to inform their practice, trusted because they knew them to be rigorously developed.

*"If an App has got some backing from a recognised guideline. NICE for example, then that's going to make you more confident about recommending and using it I think. So that would be the main thing is the confidence and remembering that it exists in amongst the plethora of other Apps really. So if there was one that we knew was recommended by a reliable source then that would be helpful."* Commissioner, participant 12.

*"It* [policy or regulations] *would have an influence . . . if you were to take NICE. . .the stronger the recommendations . . . the further up the* [priority] *list. . . and they* [large charities, e.g., Stroke Association] *will advocate them on behalf of patients."* Physiotherapist, participant 6.

However, from a commissioning perspective, embedding evidence-based tools into routine practice was seen as complex in UK settings. One commissioner explained that the challenges to implementing usable digital tools begin at the procurement stage and are often overlooked as they are subsequently deployed.

*"[On the impact of policy and evidence-based guidelines for implementing digital tools] . . . the NHSX and NHS Digital have done the digital assessment questionnaire and more recently the DTAC* [Digital Technology Assessment Criteria] *. . . that really does bring together the NICE guidelines, the evidence around clinical assurance, the regulatory things you need to consider when it comes to data use and to cybersecurity. There is an awful lot that you need to consider and it* [digital tools] *probably really doesn't get properly embedded when procuring a technology that's fit for purpose."* Commissioner, participant 2.

**Barrier 2: Engagement of healthcare professionals with tools.**   Participants mentioned a perceived lack of engagement by healthcare professionals as a significant barrier to implementation of digital tools in practice. These included the nurses and physiotherapist, who felt this lack of engagement was largely because professionals felt unable to use the tools. GPs perceived the lack of engagement to be more likely due to lack of time or resistance to change. However, a commissioner and a GP felt that a whole multidisciplinary team commitment to digital implementation was crucial to ensure long-term engagement by healthcare professionals.

*"The conversation needs to be part of the blueprint. You need that high-level engagement from your steering group, or your project board, and then what you do is that you set the tone. These are the workshops that . . . will have certain representation from each one of those."* Commissioner, participant 2.

*"I think the people who they're the people who go first with lots of different things will probably get on it straight away and try and push it and then the other 80% will be oh yes we'll get to it at some point."* GP, participant 13.

*". . .the first big shift to digital that we've probably had as a PCN which has been met with variable response from patients and GPs and staff."* GP, participant 15.

From a commissioner's view, a potential solution to promote early engagement would be to train and foster familiarity within clinical teams, in the LTC need(s), and in understanding of how particular digital tools work to support their LTC patients.

*"I'm really lucky here that I have a team, all of us come under the same portfolio and we have worked out our operating journey between us . . . at least if we understand that we can influence how it needs to work when the change leads come and engage our services."* Commissioner, participant 2.

**Facilitator 3: Voluntary sector involvement.** Frequently mentioned examples of voluntary sector providers, who either endorsed, developed, or provided access to digital tools used by GPs and healthcare professionals, included the British Heart Foundation, Asthma UK, Diabetes UK, British Lung Foundation, Kidney Research UK, and the Stroke Association. The British Heart Foundation was the provider most often mentioned for offering digital tools. Some interviewees described using and trusting these provider's websites to source support for specific conditions as they believed their information to be reliable and accessible. One GP explained:

> "*There could be almost any number of charities where it would add even more weight. You've mentioned Age UK but I think perhaps even British Heart Foundation, various other ones that are catering for some of the long term conditions. So I think the more professional bodies and charities that are giving the same message the better really.*" GP, participant 9.

Most healthcare professionals perceived endorsement, or development by, voluntary care providers, as an important factor facilitating the implementation of digital tools in practice. Because they worked with individuals with specific conditions, voluntary care providers were viewed as staff who could develop expert knowledge about particular apps, and therefore, promote and support their use with individuals with this condition.

**Barrier 3: Professionals investing time in digital tools.** Lack of time was reported as an important barrier to using digital tools in their practice by nearly all participants. They felt time was required to understand and adopt digital tools, including time needed to research the digital tool, time to make sense of it, time for training, and time to determine if the digital tool was usable in the way they needed it to be, and their patients. Healthcare professionals felt that even if a digital tool was evidence-based, if there was insufficient time to invest in preparing to use it, it would not be implemented. Interviewees clearly felt bad about this, but in the words of one GP:

> "*I sound really awful, and I don't mean to. . .so, I have no time. . .just no time. That is the danger.*" GP, participant 8.

> "*We haven't got loads of time that's the thing to even switch off our phones we're contracted from 8.30am until 6pm so we literally have no down time. So short supported and acknowledgement that the implementation takes time.*" GP, participant 14.

Lack of time was a barrier for all healthcare professionals, in as far as they had no time to train, use, or teach patients to use the tools. Managers lacked time to deliver staff training and education. Participants also felt that the time demanded of patients in learning to use, and then using the tools for their own self-management was a major barrier.

## Cognitive participation: Enrolling and engaging individuals with new digital tools

**Barrier 4: Integration into current IT systems.** Ten participants discussed ways in which digital tools could be integrated into existing health service information technology (IT) systems, thus facilitating their implementation into practice. One of the commissioners felt that individual digital tools would not be useful unless they were integrated into the service's IT infrastructure. GPs tended to feel that whatever the digital tool was, it would need to be integrated with the current primary healthcare IT systems. Systems for making referrals, managing patient records, and follow-ups could be incorporated. Some of the GPs specifically mentioned

the benefits of Accurx, an IT system that emerged pre-COVID-19. This provides a platform for professionals and patients to communicate with each other. A GP and a commissioner commented on the efficiency and added functionality of incorporating digital tools into existing IT platforms:

*"So that would make it a lot easier to use from our point of view, rather than having a completely different system that you had to put the patients details in to, and so on. Once you are logged into the system, it would be nice not to have to log into another system to get it all working. It would then be good if it could record the results in the clinical system as well."* GP, participant 3.

*"What do you want the patients to do? How do you want them to be a partner in the care that they either track or that they help themselves with? How is that going to feedback to you? Do you want that as a standalone tool? Why bother if it's a standalone tool?"* Commissioner, participant 2.

**Facilitator 4: GP network involvement.**   Nine GP's described GP networks as facilitating the implementation of digital tools, both in terms of coherence (i.e., making sense of the digital tool, for operationalising into clinical/community practice), but also cognitive participation and reflective monitoring (i.e., the appraising or evaluating the effect of the digital tool). GP networks were described as giving practitioners timely feedback on current issues in clinical practice, including the use of digital tools. GPs suggested that the COVID-19 pandemic had generated both greater need and also opportunity, to share information with other GPs. For example, one GP said:

*'There are lots of Facebook groups that have had huge coverage of GPs. Things like Resilient GP, [Teko] GP group. These all reach a broad forum of GPs."* GP, participant 15.

Our GP's believed that professional networks would enable a digital tool to either become widely promoted and embedded in practice or quickly rejected. Commissioners, nurses and the physiotherapist did not appear to have access to similar networks, but some were aware of the GP networks and appreciated their utility:

*"WhatsApp and things will come up. . .they'll (GPs) say 'Oh, have you seen this link to . . ..? This is useful. . ."* Nurse, participant 5.

**Facilitator 5: Digital leads or champions.**   Participants (including GPs, a nurse, physiotherapist, and the commissioners) considered that having a digital lead for their service as a champion, or advocate, would ensure all stakeholders are involved in the implementation of a tool. They suggested champions for particular digital tools, who would promote and assist with use for professionals and patients of the new digital tool. Suggestions for who this facilitator should be included a lead LTC nurse, a GP, or healthcare assistants (HCA). One of the commissioners describes the valuable input of a digital advisor:

*"There's a change management piece that comes with it and then there's an ongoing maybe resource allocation that could come with that. So, for instance the industry provider that I work with around long-term conditions provides us with a digital health adviser at the same time that helps with our onboarding and also provides knowledge exchange activities for new*

*care coordinators that are taken on in public health and within primary care networks."*
Commissioner, participant 2.

**Facilitator 6: Changing attitudes post-COVID-19.** COVID-19 was mentioned by all participants as a factor that had changed their perspective on use of digital technology in healthcare. The service-wide implementation of new digital technologies, and the speed at which they were adopted during the pandemic, were key factors in changing healthcare professional's perspectives to one where they viewed digital tools as viable and often critical option. Interestingly, the range of technologies that healthcare professionals and commissioners perceived as 'digital tools' facilitating self-management for people living with LTCs, was wider than had originally been conceived, when the study was designed. This included technologies that were platforms, rather than specific tools. For example, Accurx, a software provider offering an online platform that allowed them to communicate with their patients via SMS, email or the NHS app was valued by GPs and healthcare professionals. COVID-19 appeared to have changed attitudes and accelerated the 'digital' communication pathways, in particular Accurx and GP online forums. For instance, one GP said:

*"So that's the other thing that would . . ., and again during COVID, this was borne out that a lot of things got adopted in fairly short order. . .by word of mouth really between GPs on GP forums and things. So yes, whether that'd be through in-house groups like our own WhatsApp. . .Facebook groups like GP Survival and GP Partners, and these sorts of things. Things that are on there that people are using and finding good that's often how you find out about these things."* GP, participant 12.

Although COVID-19 changed attitudes for accepting digital tools, half of the participants expressed concerns about the security of patient data when using digital tools. Participants were concerned about how safe a digital tool was, in terms of data protection and privacy, and how likely data were to be accessed by external parties, including partner organisations or cyber criminals. Some also reflected that their patients were also concerned about having their personal data accessed.

## Collective action: What needs to be done to enact new digital tools

**Barrier 5: Digital literacy.** Participants perceived the lack of digital literacy of both healthcare providers, and patients, as barriers to use of online technology. Healthcare professionals need a degree of confidence in dealing with digital interventions to use them in their care provision, and patients would need to be digitally literate to engage with digital tools and services. Interviewees felt that groups of healthcare providers and patients had members with limited digital literacy. One on the commissioners describes four levels of digital literacy:

*"I think there are four levels of digital literacy. So one is that they just don't have the access to the technology or the skillset, they have access to the technology but don't have the skillset, they have the skillset but no access to the technology or they have access to the technology and the skillsets."* Commissioner, participant 2.

*"It's quite variable what patients will and won't respond to so some patients are highly digitally literate and more than happy and comfortable to go through digital media and others really aren't."* GP, participant 4.

"*I mean obviously there's a mixture but I think on the whole the younger demographic is probably more open to using Apps or web-based technology than the 65+ age group. One of the issues we sometimes have is there's a fair proportion of that age group which don't even have a mobile phone or a smartphone they have an older style phone so they can't actually get web-based technology or even text messaging through that. So that's sometimes a barrier in itself.*" GP, participant 13.

**Barrier 6: Patient characteristics.** All participants made reference to patient characteristics as perceived barriers to using digital tools in practice. Healthcare professional's perceptions of characteristics included patients being unable to use digital tools, which six participants believed was due to their inability to use the digital tool (previously discussed as a lack of digital literacy). Another was the belief that patients' conditions and priorities made them less likely to use a digital tool. For example, patients living with chronic pain, and the visually impaired were suggested as being less likely to engage in mobile or computer-based tools.

"*There are certain things that you wouldn't want to send it to someone say for example who is palliative who was a paraplegic, it would be insensitive.*" GP, participant 8.

Affordability, particular for new models or versions of technologies, was also an issue for some patients. For instance, one GP reported:

"*. . .the people we really struggled with showing it to are people of lower socioeconomic or educational background I should say, lower educational background above 30 haven't done so well. Make that no secondary education and above 60 and you're really stuffed I think.*" GP, participant 1.

"*It depends on the individual, depends on their long-term health condition and the person they are as well in terms of some people are very self . . . can self-manage and feel quite confident doing that whereas others if they are still quite anxious. If it's quite a new diagnosis I think that can. . . It's just having that interaction with someone as a person rather than just on an app. Whereas other people are quite happy to do it by an App.*" Commissioner, participant 10.

Ultimately, a digital tool would need to be technologically and culturally accessible ensuring inclusion of patients irrespective of needs, diagnoses, age, gender, and not too complex for them to understand, or use.

"*I think trying to design one thing for everybody is a route for failure and we're in danger of throwing out something that does add value just because it's not perfect for everyone.*" Physiotherapist, participant 6.

Motivation was described by the participants as being an important factor in whether or not patients engaged with, and maintained engagement with, digital tools. Healthcare professionals recognised that some patients had more to overcome, no matter how motivated they were to exercise. The only solution they had to combat the lack of motivation, was to pass on information about how much benefit they might feel from exercising, and engaging with digital tools to assist with this:

*"I think the other problem can be maintaining . . . if you start on some enthusiastic programme to keep fit or eat well you get to a point at which you hit a bit of a wall, and how you get someone through that. . . liven up the digital technology in some way. I'm no expert, but I think that would be an issue. If it could be made in some way to maintain interest . . . but quite how you do that I don't know."* Nurse, participant 5.

*"Other limitations I guess just motivation in general. The same for everyone on the planet right I guess but motivation, motivating our patients to do that. Certainly, advanced stage CKD they are symptomatic so usually they eat less because they feel nauseous a lot of the time and they generally have less energy. They are often anaemic. So motivating that patient base to exercise is very, very difficult. I think unless you make it very clear to them about the potential benefits of it and that's I guess where the education comes in."* GP, participant 11.

Participants also cited the importance of acknowledging that the symptoms of some conditions made it more difficult to engage with and maintain physical activity to help manage their condition(s).

## Reflective monitoring: Informal and formal appraisal of digital tools over time

**Barrier 7: Accountability and monitoring.**   Most participants felt there needed to be a way of ensuring accountability to support the implementation of digital tools. If the technology was for patient self-management, there would need to be a professional checking the system to ensure that patient symptoms (e.g., blood samples, blood pressure) were being monitored in case the patient's condition deteriorated. They also believed that if a patient was using a digital tool without any healthcare professional guidance or support, that this could compromise the patient's care. For example, a commissioner and GP described:

*"That is what we are trying to make sure that we continue both by virtual nudges, but also about making sure that it's properly embedded to reviews, advice and guidance so that we're keeping patients active using those tools."* Commissioner, participant 2.

*"The only problem really, I found with the text reminders for oxygen readings, and so on, for COVID patients was it's a bit unclear what we did with them out-of-hours. So did they get a text over the weekend, or on bank holidays, and things like that, and if so, who dealt with those. What would happen they sat there in the ether, going to no one in particular. Those were the main issues with that one."* GP, participant 12.

**Barrier 8: Long-term investment.**   An important barrier to sustaining use of a digital tool mentioned by health care providers was the lack of long-term investment in digital tools. If use of these digital tools was not centrally funded, GPs were not in a position to fund it themselves. Mechanisms for funding by the CCGs was complex. In relation to this, one of the commissioners discussed making bids for funding for digital tools:

*". . .so from a commissioning perspective we write service specifications. So, we usually go 'here's what we are looking for you to achieve' and we provide the bids for that and we provide the funding should they be successful."* Commissioner, participant 2.

*"The reason we use Accurx, is because that has been funded, so I guess we are basing what services we use on whether, or not we need to fund them ourselves. . ."* GP, participant 3.

". . .in the previous practice where I worked, we had the telehealth that was through the CCG at the time. That was a CCG initiative in that area, but it was a different area to where I work now. Yes they definitely pushed that quite hard and funded it, so that made a big difference obviously in us implementing it." GP, participant 4.

**Facilitator 7: Positive feedback from professionals and patients.** Feedback suggesting that a tool is evidence-based, time saving, and has benefits for patients and clinicians, was reported by nearly all participants as a key facilitating factor in implementing digital tools in practice. Feedback was also deemed crucial for clinical implementation and long-term, continual monitoring (promoting quality assurance and adverse event monitoring). Positive feedback aligns to the NPT's reflective monitoring. Regular provision of this could help integrate a digital tool into routine practice and culture, and subsequently promote health professional coherence. As one professional explained how they provide positive feedback, whilst promoting patient education:

". . . we have a link to a paediatric app that is fantastic . . . I text that out to parents all the time [the link to the App] . . . you can select a condition and the idea is to feedback and educate patients . . . when to present and when not to present." GP, participant 8.

If there was evidence that a digital tool had usability, benefitted patient care and self-management, and that the tool was increasing capacity for clinicians, these were all seen as enabling factors. This feedback would have to come from the healthcare professionals using and endorsing the digital tool at a local level, from the service and/or commissioners at a regional level, and the voluntary sector and charities at a national level. Saving time and benefiting patients and healthcare professionals was particularly important feedback, early in the implementation process to support continued use of the digital tool.

## Discussion

This study explored factors hindering and facilitating the implementation of digital tools for self-managing and maintaining physical activity, for those living with LTCs, from the perspectives of NHS healthcare professionals and commissioners. Fifteen professionals were interviewed, including ten GPs, two nurses, a physiotherapist, and two commissioners. We found that many existing digital tools to support physical activity are not implemented successfully into clinical practice; this is partly attributable to digital technologies not always accounting for the complexities of clinical practice.

Framed using the NPT's [28] four domains, our findings suggest that NHS healthcare professionals commonly recommend digital tools to help people with LTCs self-manage their condition(s). The COVID-19 pandemic has increased the number of such recommendations. However, we found the implementation and uptake of digital tools are contingent on their usability/acceptability, condition-specificity, endorsement, trustworthiness, including perceptions of data security and of the soundness of the evidence-base, and digital literacy. From a LTC patient perspective, Ward and colleagues' [36] review of use of telehealth in primary care, concurs with our findings of the importance of 'trustworthiness' and 'digital literacy' to successful implementation. Telehealth was reportedly beneficial when there was a 'pre-existing patient-professional relationship' and when patients were digitally capable of using telehealth at home (e.g., remote blood pressure cuffs/glucose monitors). Drawbacks included the need for 'close monitoring' by professionals (particularly for complex comorbidities), which relates

to time and additional support for LTC patients unfamiliar with telehealth [36]. Digital tools are being rapidly adopted in healthcare services worldwide. This study highlights limitations in existing digitals tools and factors influencing their use in supporting LTC self-management in the NHS.

The COVID-19 pandemic led to the rapid adoption of digital technologies in the NHS, requiring healthcare professionals to adopt innovative methods of service delivery [37]. Our interviews clearly demonstrate changed healthcare professional attitudes to digital technologies post COVID-19. For our healthcare professionals and commissioners, the main reasons for accepting and engaging in using and recommending the use of digital tools were i) the patient/clinical needs, ii) opportunity for innovation in healthcare, and particularly for GPs, iii) to share information with peers nationwide. Van der Ham et al. [38] found evidence of a general pressure on healthcare professionals from patients, managers and colleagues to use digital tools. Digital tools are perceived as being money-saving and efficient, regardless of whether they are found to be useful or easy to use. Positive attitudes towards use of digital tools were promoted by exchanging experiences via professional networks, active use in practice, and formal discussions within multidisciplinary teams and patient/charity organisations [38]. Our interviews showed GP networks to be highly influential fora for exchanging information about digital tools, indicating the best tools and those that were not so helpful. Like van der Ham's [38], our data suggest that the majority of healthcare professionals are positive about digital tools. Commissioners and LTC patients were also predominantly positive. Van der Ham and colleagues' [38] data was cross-sectional survey data, so provided no insights into reasons for this positive attitude. This gap is filled by our participant's observations that these tools could address patient and clinical needs, and that they offer opportunity for innovation in healthcare (e-consulting for example) and information sharing with colleagues. There was a perception that these innovations were rapidly integrated into existing health systems because of the versatility of digital tools. However, a negative of this rapid uptake, was the potential for security risks and data breach for our healthcare professionals.

Condition-specificity was deemed crucial for supporting implementation of digital tools because it allowed users to make sense of the digital tools, thus achieving a sense of coherence in line with NPT. A digital tool must be tailored to the needs of the specific and primary condition. For example, physical activity guidance will differ between people with COPD and diabetes as their primary condition, partly due to different pathologies, exercise tolerance, and dietary needs [5, 13]. The World Health Organisation's (WHO) physical activity guidelines now provides recommendations for people with LTCs, whilst also recognising the role of healthcare professionals in providing tailored advice [5]. WHO's guidelines suggest similar guidance can be given across LTCs, but specific advice on where to start and how to progress (i.e., frequency, intensity and duration) requires healthcare professional input.

Many existing digital tools to support people with LTCs staying active, focus on single LTCs without considering additional comorbidities [21]. In the UK alone, around 54% of over 65-year-olds are now living with multiple LTCs [39]. A new digital health platform, ProACT, is being co-designed with older adults with multiple LTCs in Ireland and Belgium [27] to support self-management on a single platform, offering symptom monitoring, condition-specific education and data sharing with health services. Over a 12-month trial, patient engagement was found to be high (78%), mainly because patients saw value and benefit in the platform, but also the usability and low-burden associated with a self-reporting and monitoring. Health professionals were involved in designing the digital platform, reflecting the need expressed by our healthcare professionals and commissioners, and previous work [40], for digital tools to be usable and acceptable, and for them to be endorsed by peers. All of which were present in the design and deployment of ProACT [27]. Similarly, the It's LiFe digital tool, was developed

with people living COPD and type II diabetes, and effective in complementing a self-management programme to increase physical activity up to 3 months post-intervention [13]. Technology that is well-designed from a usability perspective can increase the utility of the technology/ digital tool, reduce potential error, enhance user acceptance, and consequently improve productivity [40].

A recent scoping review of digital technologies for self-managing LTCs in children [41], identified three factors important for adoption into practice: i) feasibility and acceptability, ii) usability (including aesthetics, ease of use, and device synchronisation), and iii) promoting adherence and improving self-management skills. Research suggested that to achieve these technologies need to be evidence-based [42, 43]. NICE was a source of evidence trusted by healthcare professionals, which because of it's perceived reliability and rigour, gave them confidence in advocating tools to their patients. In common with implementation of any healthcare innovation, lack of time for learning about and tutoring patients in use of digital tools was a major barrier. One widely experienced example of this is the introduction of virtual consultations. Despite the potential for digital tools like this to save time, examples in triage suggest that telephone and virtual-first consultations can increase clinician workload [44, 45]. These considerations are particularly relevant for commissioners, who need time to evaluate the trade-offs involved in procuring and implementing digital tools. With NHS clinicians facing increasing time pressures, and the majority of smartphone apps for physical activity promotion lacking an evidence-base [46], it is a major challenge for clinicians to identify and adopt new apps that are not already established.

Condition-related characteristics will sometimes endorse digital tools, adding to their value for users by including condition-specific advice, evidence-based information, and approval from professionals as trusted authorities. Alongside professional networks, our participants saw charity involvement as crucial in helping patients and professionals to make sense (i.e., attain coherence) of digital tools, and engage (i.e., participate cognitively) with them. Charities in this way represent and compliment peer endorsement which is known to be effective in physical activity promotion [47] and in reassuring patients as to the quality of digital interventions [48]. The peer endorsement mentioned commonly in our interviews was that experienced by GPs in their online clinical networks. Evidence exists that clinical networks can be effective pathways for quality improvement, particularly in service delivery, including supporting adherence to clinical guidelines and adoption of clinical tools [49], because they increase cognitive participation. This will also apply to implementation and use of digital tools. This suggests that promotion of new digital tools would be enhanced by integration into professional networks.

Engagement with, or cognitive participation in, digital tools appeared to be contingent on the healthcare professional's capacity to make use of the technology, with a lack of time and resistance to change being the main barriers preventing engagement with digital tools. This suggests that engagement might be enhanced by integration into existing IT systems used routinely by healthcare professionals to minimise the demand on their time, and also by exposure to endorsements from champions or those in their clinical networks. Integration into existing IT systems is an obvious solution, as it offers efficiency, and may reduce the time required by LTC consultations, which can involve reviewing multiple conditions. The potential benefits of having a digital lead, or 'champion' were also obvious to our healthcare professionals and commissioners. Digital champions were integral to the successful implementation of a new digital communication app, Pulsara™, designed for the secure sharing of patient details, symptoms, care times and monitoring [50]. Digital champions of this app supported research staff in delivering formal and ad hoc training, alongside demonstrating the app to colleagues. Our interviewees felt that digital leads should have knowledge of common LTCs and self-

management, both of which would be necessary to support peers in training. Our study participants made the point that there would need to be flexibility in within a job if being a digital champion was to be managed alongside other clinical/administrative roles. Recognition of, and adaptation to, the contextual workplace demands on healthcare staff is clearly important to successful implementation in care for those living with LTCs.

Effective collective action to implement new digital tools was found to be influenced by digital literacy, which mainly concerned the ability to use smartphones, but also broader technologies, such as IT systems and internet use. Both patients and professionals may support in this area. This is an issue most readily addressed by digital champions [50]. The rapid uptake of digital technologies in healthcare has raised concerns that they may increase health inequalities, with digital alternatives not accessible or used by certain groups [37]. This was highlighted by our two commissioners, who based on their interviews, were digitally literate and aware of health inequalities in the region. Patient preferences, age, socio-economic status, and the nature of their condition (e.g., sight impairment) were important determinants of their capacity to engage in, and utilise digital tools, particularly smartphone apps. Doyle et al. [27] previously found that digital tools had to be simple to use and included technical support if they were to be used by older adults, for example. Given that self-management refers to an individual's ability to manage their condition(s) in daily life [7], it makes sense that patients and healthcare professionals are involved in the design and delivery of digital tools [40] to promote LTC self-management because their experience of everyday life needs to be accommodated by the tool design. Existing examples of successful digital platforms [27] and apps [13, 50] have involved patients and professionals across design, delivery and evaluation phases.

Our data suggest that reflective monitoring in the form of systems and accountability structures is an important part of successful implementation of digital tools for LTC self-management. Systems for monitoring use of digital tools, needs to include patients being signposted to endorsed and evidence-based digital tools, remote monitoring and support for symptom management, and timely and responsive decision-making. Based on an Nuffield Trust report [37], these is still not much in evidence in the UK. Commissioners can influence policy in this regard, working nationally and/or regionally with national bodies such as NHS England or NHS Digital to secure long-term investment, with which to procure and integrate digital tools within health service infrastructure. COVID-19 may have created a 'teachable moment' for such long-term planning [37], and implementation of the UK NHS' Long Term Plan [51], which commits to promote the mainstream implementation of digital care across the NHS. Our findings suggest that positive feedback is required at both individual healthcare professional and service level, prior to, and early in, implementation of digital tools. The bottom line for healthcare professionals in accepting or rejecting a digital tool for use in their practice is, 'does it save time?', 'does it benefit patients?', and 'does it work?'.

## Strengths and limitations

Strengths of this qualitative study were the diversity of our healthcare professional stakeholders including GPs, nurses and commissioners. This diversity of experience was reflected in their insights into experiences of using digital tools in NHS settings during, and beyond COVID-19. The application of the NPT implementation framework structured the learning from the participants into a series of clear recommendations about how digital tools could be implemented successfully and impact, and be adapted by stakeholders. This paper therefore, contributes to advancing practice in this area, and the UK NHS' Long Term Plan [51] goals relating to reducing potential wastage of time and resources, by a review of considerations to be taken into account when a digital tool or resource is procured. When interpreting the findings, it is

important to be aware that most participants were GPs in primary care in southern England. This may have influenced themes relating to time and workload, specific to their work setting, it's culture in respect of the acceptance of and engagement with digital tools by their NHS management, available funding, infrastructure, and related demands. Another limitation is that those GPs who were prepared to take part in this research may only represent those healthcare professionals who are already engaged in new technologies. Equally, we anticipated difficulty in recruiting commissioners, and thus adopted convenience sampling to recruit two from neighbouring counties. Accepting these limitations, our group of participants provided insights into issues that clearly impact their working relationship with digital tools, and which are likely to have relevance for any healthcare professional involved in their implementation.

## Implications for policy, practice and future research

Digital tools are a significant part of our daily lives, and in 'Digital Transformation', are pivotal to the UK NHS' Long Term Plan [51]. They provide multiple pathways for people with LTCs to access healthcare, maintain physical activity, and self-manage conditions, but as our findings suggest, are not currently being implemented effectively in the NHS. We suggest that our findings have direct implications for three different domains:

- Strategic policy—the NHS, commissioning groups and other national health organisations would do well to ensure that when procuring new digital tools, consideration is given to sourcing services that are accessible to and usable by both healthcare professionals and patients, particularly in relation to complex and multiple LTCs. This includes sourcing evidence-based technologies that are supported by national charities or other voluntary sector providers. Finally, integration of into existing health service IT systems will enable more successful implementation.

- Healthcare practice–professionals need to be supported to engage with digital tools if they are to become routine within healthcare practice. At practice level, existing staff practice in using their clinical networks to review and endorse digital tools can be encouraged. At a leadership level, managers can support staff in training and education, using existing digital tools within their service. This includes supporting staff to become digital champions, to advocate and promote specific digital tools. We are mindful of existing workload and cost issues in suggesting this.

- Future research–should engage healthcare professionals across different staff levels, years of practice, and regions, to increase the diversity of insights into use of digital tools and their motivations for doing so, but also to explore the specific educational needs of professionals in their implementation. Future intelligent tools that can be tailored and individualised for people with different and changing needs, due to multimorbidity are also warranted.

## Conclusions

This study provides insights into healthcare professional and commissioner's issues with using digital tools in the NHS to support people with LTCs to self-manage their conditions. Reasons for digital tools not being consistently implemented, related to a lack of condition-specificity and accessibility for different patient groups, poor usability and acceptability, absence of an established an evidence-base, and the need for involvement and endorsement from voluntary sector and professional networks. To promote clinical engagement and implementation at scale, our professionals felt digital tools must be integrated into existing health systems, be

championed by professionals, designed around patient and professional digital literacy, and monitored and invested in long-term. Only through these actions would the ambition of the NHS Long-term Plan for digital transformation of healthcare be fully realised.

## Supporting information

**S1 Fig. Interview topic guide.**
(TIF)

## Acknowledgments

We would like to thank members of the wider MOTH team and the National Institute for Health Research (NIHR) Applied Research Collaboration (ARC) Wessex staff, for their helpful involvement in the programme and administrative support, particularly during the COVID-19 pandemic.

## Author Contributions

**Conceptualization:** Paul Clarkson, Paul E. Muckelt, Euan Sadler, Suzanne McDonough, Mary Barker.

**Data curation:** Paul Clarkson, Paul E. Muckelt.

**Formal analysis:** James P. Gavin, Paul E. Muckelt, Rachael Eckford, Euan Sadler, Suzanne McDonough, Mary Barker.

**Funding acquisition:** Suzanne McDonough, Mary Barker.

**Investigation:** Paul Clarkson, Euan Sadler, Suzanne McDonough, Mary Barker.

**Methodology:** Euan Sadler, Suzanne McDonough, Mary Barker.

**Project administration:** James P. Gavin, Paul Clarkson, Paul E. Muckelt, Euan Sadler, Suzanne McDonough, Mary Barker.

**Supervision:** James P. Gavin, Euan Sadler, Mary Barker.

**Writing – original draft:** James P. Gavin.

**Writing – review & editing:** James P. Gavin, Rachael Eckford, Euan Sadler, Suzanne McDonough, Mary Barker.

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
