## [Decision Letter · Decision Letter 0]

19 Feb 2024

PONE-D-23-37603Healthcare professional and commissioners’ perspectives on the factors facilitating and hindering the implementation of digital tools for self-management of long-term conditions within UK healthcare pathways.PLOS ONE

Dear Dr. Gavin,

Thank you for submitting your manuscript to PLOS ONE. After careful consideration, we feel that it has merit but does not fully meet PLOS ONE’s publication criteria as it currently stands. Therefore, we invite you to submit a revised version of the manuscript that addresses the points raised during the review process.

We look forward to receiving your revised manuscript.

Kind regards,

Michael Thomas Lawless, Ph.D.

Academic Editor

PLOS ONE

[We would like to thank members of the wider MOTH team and the National Institute for Health Research (NIHR) Applied Research Collaboration (ARC) Wessex staff, for their helpful involvement in the programme and administrative support, particularly during the COVID-19 pandemic.]

 [This study was funded by the National Institute for Health Research (NIHR) Applied Research Collaboration (ARC) Wessex. The views expressed in this publication are those of the author(s) and not necessarily those of the National Institute for Health and Care Research or the Department of Health and Social Care (https://www.arc-wx.nihr.ac.uk/).

The funders had no role in study design, data collection and analysis, decision to publish, or preparation of the manuscript.]

Reviewers' comments:

Reviewer's Responses to Questions

**Comments to the Author**

1. Is the manuscript technically sound, and do the data support the conclusions?

Reviewer #1: Partly

Reviewer #2: Yes

2. Has the statistical analysis been performed appropriately and rigorously? 

Reviewer #1: N/A

Reviewer #2: N/A

3. Have the authors made all data underlying the findings in their manuscript fully available?

Reviewer #1: No

Reviewer #2: No

4. Is the manuscript presented in an intelligible fashion and written in standard English?

Reviewer #1: No

Reviewer #2: No

5. Review Comments to the Author

Reviewer #1: This article reports the results of an interview study exploring the barriers and facilitators to the use of digital technology in supporting physical activity to manage LTC from the persepctive of health care professionals and comissioners.

Generally the article lacks clarity and would benefit from being more specific. For example in parts the aim appears to be the use of digital technology to promote PA in the self management of LTCs but in parts it appears the aim is to explore digital technology more generally.

Specific comments below

Abstract

1. The opening sentence appears incomplete

2. It would be helpful for non-UK audience to explain where Wessex is

3. State which population was purposively sampled and in which popualtion convenience sampling was used.

4. Make sure the conclusion alighs wihth the results e.g. the conclusion talks about assessibility but this isn't mentioned in the results.

Introduction

5. The opening sentence talks about PA 'speciically to improve symptoms'. There needs to be a few sentences explaining how PA might improve symptoms.

6. The sentence on line 92 appears incomplete.

7. How did the earlier scoping review inform this study?

Methods

8. Give examples of other health care professionals (line 123)

9. Explain more about purposive sampling e.g. which characteristics were sampled and how people were identified and approached.

10. Consider including the topic guide as supplmentary material

11. Inclusion and exclusion criteria need to be clearly stated e.g. did the interviewees have to have experience of promoting or using digtial health in LTCs

Findings - this section would benefit from greater clarity with furher development of the themes and ensuring that the themes relate to the use of digital tools in managing LTCs.

12. Sentence on lines 187-188 lacks context/ justification.

13. Table 1 needs formatting, it is unclear which participant column 6 aligns. Consider splitting column 6 into two columns one stating whch LTCs had been supported using digital technology and the other what digital technologies had been used,

14. Why is athma included in column 6 when no apps had been used?

13. In LTCs listed in column 6 I suggest removing any conditions where digital technology hadn't been used to support the condition. As it stands the table is confusing as it appears that in column 6 in some cases all LTCs that the participant supports have been included and in other cases only LTCs that the participant has supported where digital technology has been utilised.

14. Table 2 is confused and needs revising. It is unclear which domain aligns with which theme.

15.Quotes need to be chosen which accurately reflect what the authors are trying to illustrate, at present the majority of the quotes don't do this. They are either not specific enough to illustrate the theme or they appear unrelated.

16. The theme about tools being condition specific then contains a sentence about cultural specificity, this is a different theme to condition specificity and needs to be removed and placed in its own theme.

17. What is meant by functionality. It is unclear what is meant by 'digitlal tools need to be functional' - is this the best word?

18. The theme digital tools involving vulntary sectors, is this referring to exisiting tools, potential tools or visiting website? This section would beenfit from greater clarity.

19.Investing time theme - who is investing the time? This theme would benefit from more explanantion e.g. if the GP needs to invest time why do they need to? What is the purpose of investing the time?

20. It is unclear what Accurex has to do with digital tools and LTCs and self-management. This needs to be explained.

21. How is the use of NHS mail related to digtial tools for LTCs.

Discussion

22. This section would benefit from greater focus rather than re-iterating the results.

23. Explain what is meant by 'digtial divide'

24. Strengths and limitations could be expanded and better explained

25. Consider including a heading 'implications for policy and practice'

Conclusions

26. Conclusions are not clearly linked to the results.

Reviewer #2: I wish to thank the editor for their invitation to review the manuscript. This is a qualitative study discusses the challenges of integrating physical activity into clinical practice for individuals with long-term conditions (LTCs), citing barriers like access, time constraints, and transport costs. Various digital tools are proposed as solutions to overcome these barriers and support widespread implementation to promote physical activity. Through semi-structured interviews conducted in Wessex, UK, with healthcare professionals and commissioners, the study explores needs, preferences, barriers, and facilitators to implementing digital tools for LTC self-management in NHS settings. Findings reveal that current digital tools lack specificity, functionality, and evidence-base, with suggestions for integration into health service IT systems, professional networks, and the importance of digital literacy for both professionals and patients. The study underscores the importance of evidence-based, endorsed digital tools that are integrated into existing health systems for successful LTC self-management.

Overall, I found the study could be potentially valuable to the field. However, there are a few areas that I think need improvement before it can be considered for publication.

1. The manuscript needs to be heavily edited for clarity. Clear and concise writing is essential to ensure that readers can understand the research findings and their implications. For example, throughout the manuscript there is a lack of differentiation between barriers to HCPs and barriers to patients. These should be separated, to provide clarity on the distinct challenges faced by each group. Separating the barriers would enable a more focused analysis of the unique factors influencing the adoption and implementation of digital tools from both perspectives.

2. There is a lack of clarity in conceptual framework: While the manuscript mentions the use of the Consolidated Framework for Implementation Research (CFIR) and the Normalization Process Theory (NPT), it fails to provide clear explanations regarding which domains of the CFIR framework were probed during the interviews. Similarly, the Normalization Process Theory is briefly mentioned, but its application and relevance to the study are not adequately explained. It is crucial for readers to understand how these theoretical frameworks were operationalized and utilized in the study to assess barriers and facilitators to the implementation of digital tools in healthcare settings.

3. Discussion of Findings: The discussion of findings in the manuscript lacks depth. There is a need for a more thorough analysis and interpretation of the interview data, particularly in relation to the identified barriers and facilitators to the implementation of digital tools in healthcare settings. Additionally, the manuscript could benefit from a more robust comparison of findings with existing literature.

In summary, while the manuscript addresses an important topic, its current presentation lacks clarity and depth in several key areas. I believe that significant revisions are necessary to strengthen the manuscript and ensure its suitability for publication in PLOS ONE

6. PLOS authors have the option to publish the peer review history of their article (what does this mean?). If published, this will include your full peer review and any attached files.

Reviewer #1: No

Reviewer #2: No

---

## [Author Response · Author response to Decision Letter 0]

6 May 2024

Please to uploaded 'Response to Reviewers [PONE-D-23-37603]' Word document, in addition to the clean 'Manuscript' and 'Revised Manuscript with Tracked Changes'.

---

## [Editor Report · Decision Letter 1]

8 Jul 2024

Healthcare professional and commissioners’ perspectives on the factors facilitating and hindering the implementation of digital tools for self-management of long-term conditions within UK healthcare pathways.

PONE-D-23-37603R1

Dear Dr. Gavin,

We’re pleased to inform you that your manuscript has been judged scientifically suitable for publication and will be formally accepted for publication once it meets all outstanding technical requirements.

Kind regards,

Sebastian Suarez Fuller, PhD

Academic Editor

PLOS ONE

Additional Editor Comments (optional):

Thank you for your revision of your manuscript in line with peer reviewer comments and recommendations. As the original reviewers and editor are no longer available, I have reviewed this manuscript and found that all identified issues have now been addressed and your article is suitable for publication without further delay.   
---

## [Editor Report · Acceptance letter]

11 Jul 2024

PONE-D-23-37603R1 

PLOS ONE

Dear Dr. Gavin, 

I'm pleased to inform you that your manuscript has been deemed suitable for publication in PLOS ONE. Congratulations! Your manuscript is now being handed over to our production team.

Kind regards, 

on behalf of

Dr. Sebastian Suarez Fuller 

Academic Editor

PLOS ONE